# Scientometric Analysis of Diesel Pollutions in Antarctic Territories: A Review of Causes and Potential Bioremediation Approaches

**Ahmad Fareez Ahmad Roslee [1], Siti Aqlima Ahmad [1,2], Claudio Gomez-Fuentes [2,3], Noor Azmi Shaharuddin [1], Khalilah Abdul Khalil [4] and Azham Zulkharnain [5,\*]**

[1] Department of Biochemistry, Faculty of Biotechnology and Biomolecular Sciences, Universiti Putra Malaysia (UPM), Serdang 43400, Selangor, Malaysia; fareezlee@yahoo.com (A.F.A.R.); aqlima@upm.edu.my (S.A.A.); noorazmi@upm.edu.my (N.A.S.)

[2] Center for Research and Antarctic Environmental Monitoring (CIMAA), Universidad de Magallanes, Avda. Bulnes, Punta Arenas 01855, Región de Magallanes y Antártica Chilena, Chile; claudio.gomez@umag.cl

[3] Department of Chemical Engineering, Universidad de Magallanes, Avda. Bulnes, Punta Arenas 01855, Región de Magallanes y Antártica Chilena, Chile

[4] Faculty of Applied Sciences, School of Biology, Universiti Teknologi MARA, Shah Alam 40450, Selangor, Malaysia; khali552@uitm.edu.my

[5] Department of Bioscience and Engineering, College of Systems Engineering and Science, Shibaura Institute of Technology, 307 Fukasaku, Minuma-ku, Saitama 337-8570, Japan

\* Correspondence: azham@shibaura-it.ac.jp

**Abstract:** Despite the continuous enforcement of Antarctic Treaty System, ATS (1961), today Antarctica is constantly plagued by hydrocarbon pollution from both legacy and present-day wastes, especially near where anthropogenic activities are the most intense. The advances of science have led to multiple breakthroughs to bolster bioremediation techniques and revamp existing laws that prevent or limit the extent of hydrocarbon pollution in Antarctica. This review serves as the extension of collective efforts by the Antarctic communities through visual representation that summarizes decades of findings (circa 2000–2020) from various fields, pertinent to the application of microbe-mediated hydrocarbons remediation. A scientometric analysis was carried out based on indexed, scientific repositories (ScienceDirect and Scopus), encompassing various parameters, including but not limited to keywords co-occurrences, contributing countries, trends and current breakthroughs in polar researches. The emergence of keywords such as bioremediation, biosurfactants, petroleum hydrocarbons, biodiesel, metagenomics and Antarctic treaty policy portrays the dynamic shifts in Antarctic affairs during the last decades, which initially focused on exploration and resources exploitation before switching to scientific research and the more recent ecotourism. This review also presents the hydrocarbonoclastic microbes studied in the past, known and proposed metabolic pathways and genes related to hydrocarbon biodegradation as well as bacterial adaptations to low-temperature condition.

**Keywords:** hydrocarbon; bioremediation; genes; sequencing

## 1. Introduction

Antarctica has gained growing attention for its crucial role in influencing the global climate besides being an important heritage site, which has piqued scientific and public interest. In their current course, intense human activities have led to instances of anthropogenic environmental pollutions and recognition of their potential negative impacts on the Antarctic ecosystems [1,2]. The constantly freezing ambient and prolonged sun irradiation during the polar winter are among the notable causes of hydrocarbon persistence, besides acting as hurdles to the currently available bioremediation techniques [1,3,4]. Despite the comprehensive etiology on hydrocarbon pollution, especially of thermogenic sources and

the methods to address these problems in Antarctica discussed in great length hitherto, systematic analyses and visualization of existing studies are apparently scarce. In order to generate meaningful visual representation of efforts dedicated to the cause, a network analysis by retrieving accessible bibliographic information from online databases such as ScienceDirect, PMC, Web of Science and Scopus can be implemented. Hence, in this study, scientometric analyses (keywords co-occurrence, co-citation and clustering) were performed using VOSviewer software, allowing for macroscopic mapping of enormous scientific literatures exclusively linked to hydrocarbon pollutions in Antarctica published in the spans of 20 years prior (from 2000 to 2020).

This quantitative study pivots around the recurring problems of diesel pollutions in Antarctica, encompassing crucial keywords from various disciplines related to the central theme including "diesel", "Antarctica", "bioremediation", "consortium" and derivative terms of similar meaning. The accomplished objectives of this review may shed some light on the (1) current knowledges and future challenges in Antarctic diesel bioremediation, and (2) the research performance and distributions by countries and subjects.

## 2. Methodology for Materials Collection

Scientometric or bibliometric analysis (used interchangeably from this point onwards) is capable of providing both quantitative and qualitative assessment of available scientific research or review articles, books, and other publications [5]. Essentially, this method enables the extraction of omnidisciplinary scientific outputs from various publisher databases, entailing the author's topic of interest, being the application of autochthonous microbial communities in diesel clean-up efforts in Antarctica, and converting them into simpler, concise infographics.

In particular, the analysis and assessment of trends in this review were set out through data mining of the Scopus and ScienceDirect libraries. The rationale lies on the enormous, combined databases which house over tens of thousands of peer-reviewed, indexed journals from multiple fields inclusive of science and technology, arts and history, medicine and social sciences, allowing for construction of detailed meta-analysis [6]. While analyzing more than one database can provide greater data range for this study, an extra layer of precaution is highly recommended to avoid library redundancies. Therefore, a restriction on source types (limit to research articles only) was applied for false-positive mitigation. Figure 1 depicts the documentation processes prior to scientometric analysis, divided into four stages; identification, screening, eligibility and inclusion [7]. While the first two stages can be sorted based on integrated filtering tools available within the online databases, the second half of the fine-tuning stages have to be done manually in the VOSviewer bibliometric tool (version 1.6.15, Centre for Science and Technology Studies, Leiden University), and thus can be labor-intensive and time-consuming.

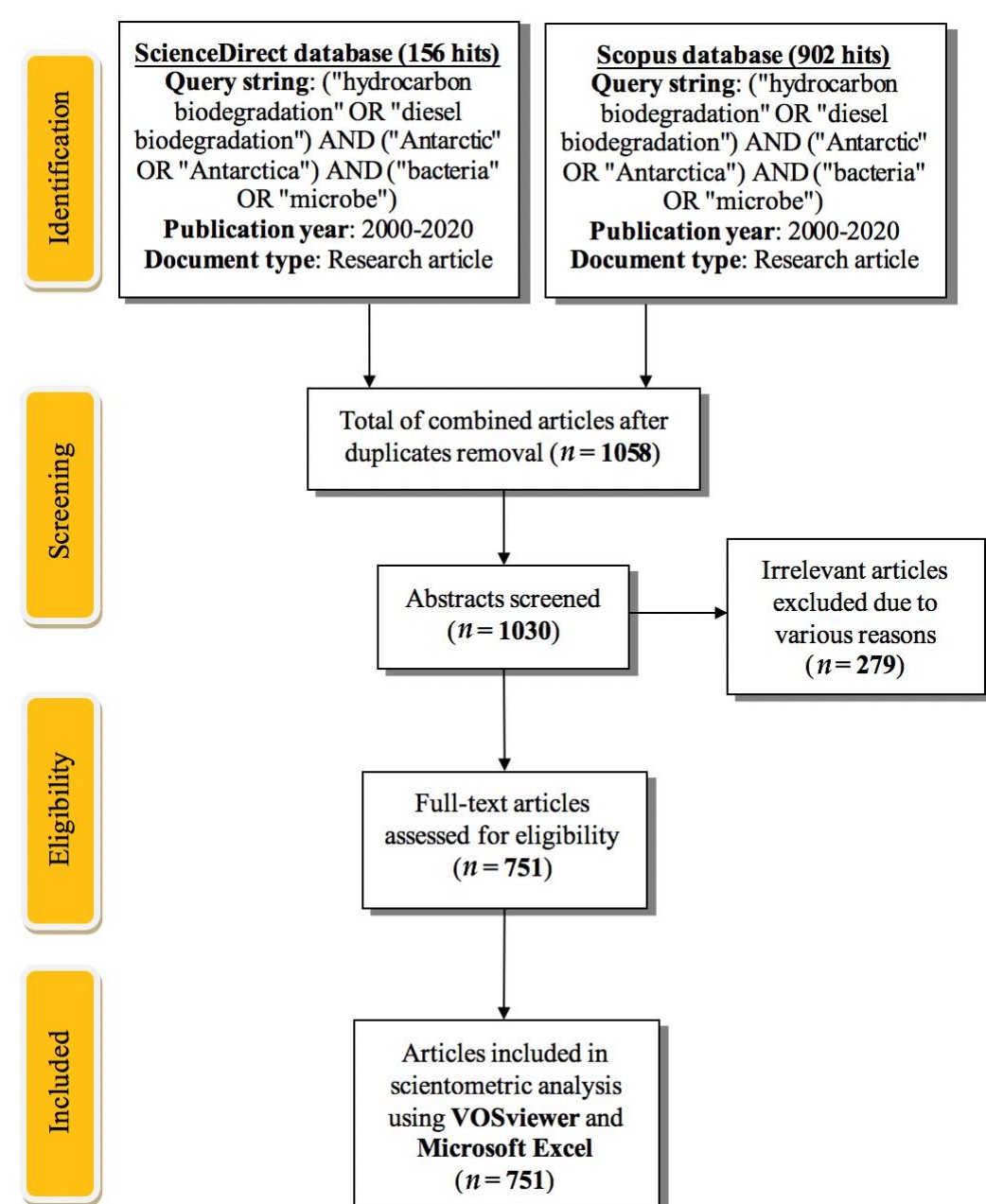

**Figure 1.** Flowchart of published materials collection from two scientific repositories in relevance to central theme of microbial-mediated hydrocarbons degradation in Antarctica.

## 3. Scientometric Analysis

The analysis focused on key indicators such as publication trends and distributions, keywords emergence and co-occurrences in the timeframe from 2000 to 2020.

### 3.1. Research Trends and Driving Factors of Hydrocarbon Pollutions in Antarctica

Hydrocarbon pollutions remain as isolated cases hitherto, happening from time to time throughout populated or frequented Antarctic areas, ranging from minor spills during vehicle refueling to potentially major incidents during ship-to-station refueling and leakage from poorly maintained pipelines or storage facilities [8]. The potential threats posed by these persistent contaminants over years have raised awareness among the polar scientists and the member nations of the Antarctic Treaty System as a whole, propelling the growth of research interests in bioremediation approaches. In Figure 2, the academic performances

pertinent to diesel bioremediation in Antarctica is shown to fluctuate steadily during the last two decades whilst registering an upward trend as backed by the cumulative outputs, depicting substantial increments from tens to hundreds of publications by the year 2020. Evidently, the increasing number of publications occurred right after the enforcement of the Protocol on Environmental Protection to the Antarctic Treaty in 1998 (also known as Madrid Protocol) [9], and has since shown periodical spikes concurrent to some major catastrophic events such as the fuel spillage from sinking vessels; in chronological order, Bahia Paraiso in 1989 (600,000 L DFA), Patriarche in 2001 (1500 L diesel), M/S Explorer in 2007 (190,000 L MGO, 24,000 L lubricant oil and 1200 L petrol), No.1 In Sung (unknown volume of fuel at time of sinking) and multiple other similar incidents involving unknown proportions of fuels [10–12]. The substantially reduced major hydrocarbon contaminations in the last decade is reflective of the significance of both governmental and nongovernmental groups (ATCPs, ATCM, CEP, CCAMLR, ASOC, SCAR, COMNAP and IAATO) in overseeing Antarctic affairs, particularly focusing on the development of safety rules and responses, and preservation of the Antarctic ecosystems [9,13].

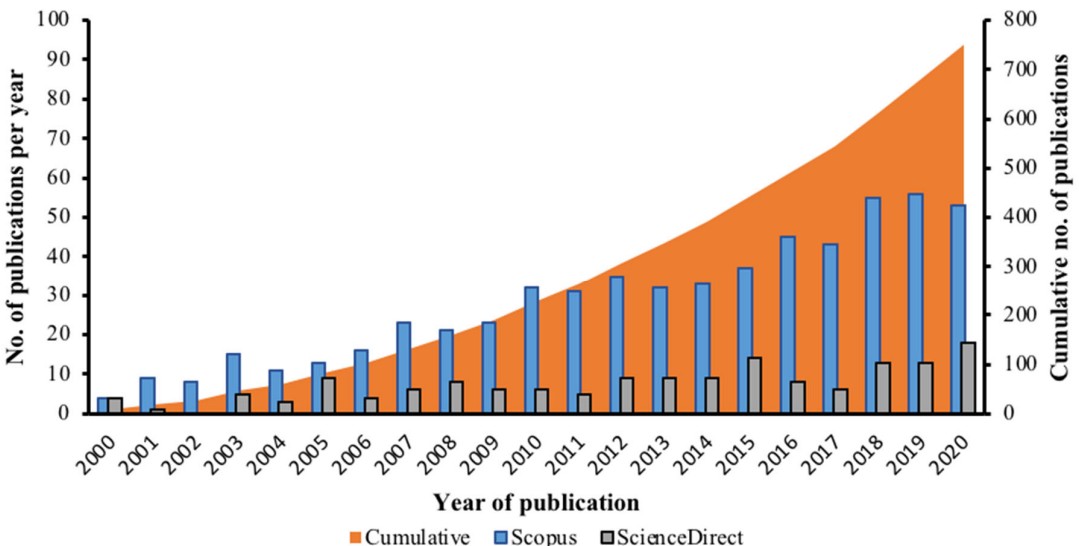

**Figure 2.** The combined number of publications from year 2000–2020 using two-level query string (("hydrocarbon biodegradation" OR "diesel biodegradation") AND ("Antarctic" OR "Antarctica")) retrieved from the ScienceDirect and Scopus databases.

Tin et al. [14] identified ecotourism as the major driver of human influx to the Antarctic continent and at the same time implied that the industry plays a significant role in contributing to the current hydrocarbon pollutions. As reported by Bender et al. [15], Antarctica has received nearly half a million visitors for the past 20 years and the figures may have been exceeded by thousands in the year 2020 (Figure 3), affiliated to the relatively recent Chinese industry as well as the existing prominent markets in the USA, UK, Germany and Australia [16]. The source of anthropogenic pollutants mainly exists in the form of thermogenic hydrocarbons linked to the high consumption of fuels from transportation (e.g., IFO, SAB, MGO and AN-8), particularly light aircraft and sea vessels. In addition, a large number of permanent scientific and military stations operating year-round are also accountable for the similitude portion of the current hydrocarbon pollution index, although most of the incidences occurred at a rather small scale and probably many more have gone unreported [17].

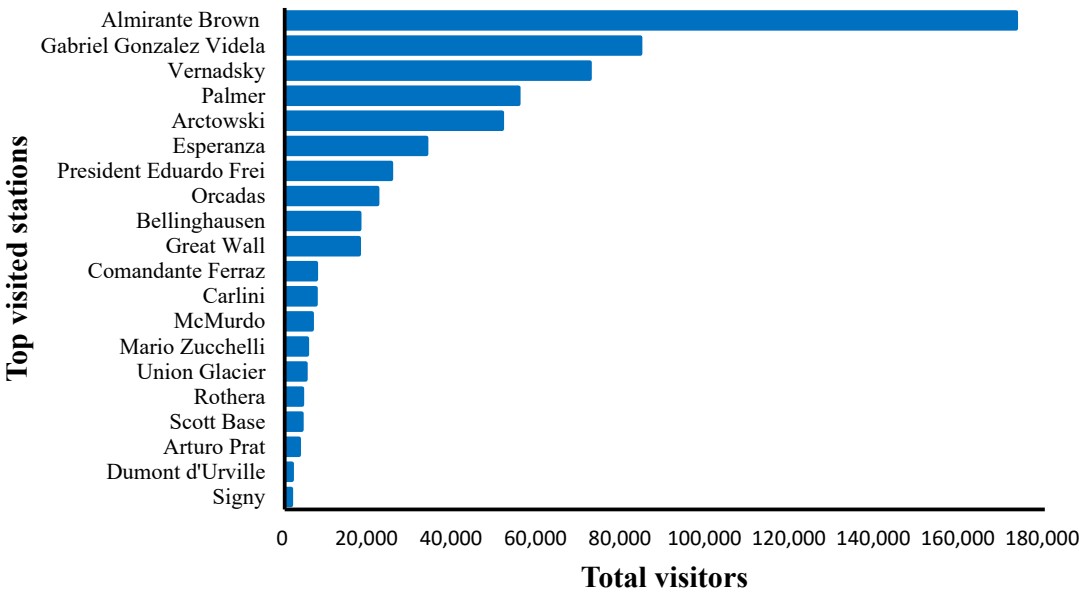

**Figure 3.** The 20 most visited permanent stations in Antarctica [16,18].

### 3.2. Contribution of Member Countries toward Antarctic Research

The bibliometric analysis using the proposed query string, (("hydrocarbon biodegradation" OR "diesel biodegradation") AND ("Antarctic" OR "Antarctica")) revealed the top 10 contributing countries in terms of publication sorted according to rank are the USA, Australia, Italy, UK, Canada, Germany, China, New Zealand, Brazil and Argentina, all of which are the consultative members of Antarctic Treaty except for Canada (Figure 4). At rank no. 14, Malaysia's contributions (as the first Asian country and 49th nonconsultative member of ATS) currently pale in comparison to other big countries with only around 2% global outputs, most probably due to the greater travel distance, lack of expertise and few facilities owned in the south pole. However, Chile, being geographically the nearest to Antarctica, was also excluded from the top 10 contenders, despite being a consultative nation and having multiple major stations built on Antarctic continent, suggesting that the number of publications do not necessarily correlate to the accessibility between researchers and the research sites.

With a closer look into the global publications, the authors were able to elaborate the critical subject areas pursued by researchers worldwide (Figure 5). The prominent subjects were grouped in the larger pie chart, followed by subsidiary studies that complement the latter either directly or indirectly. Furthermore, the cluster of significant areas of knowledge in the larger pie are regarded to be more relevant to the central theme of this review, implying that scientists from various study backgrounds, especially the environmental sciences are attentive of potential applications of microbially-mediated clean-up techniques in Antarctica.

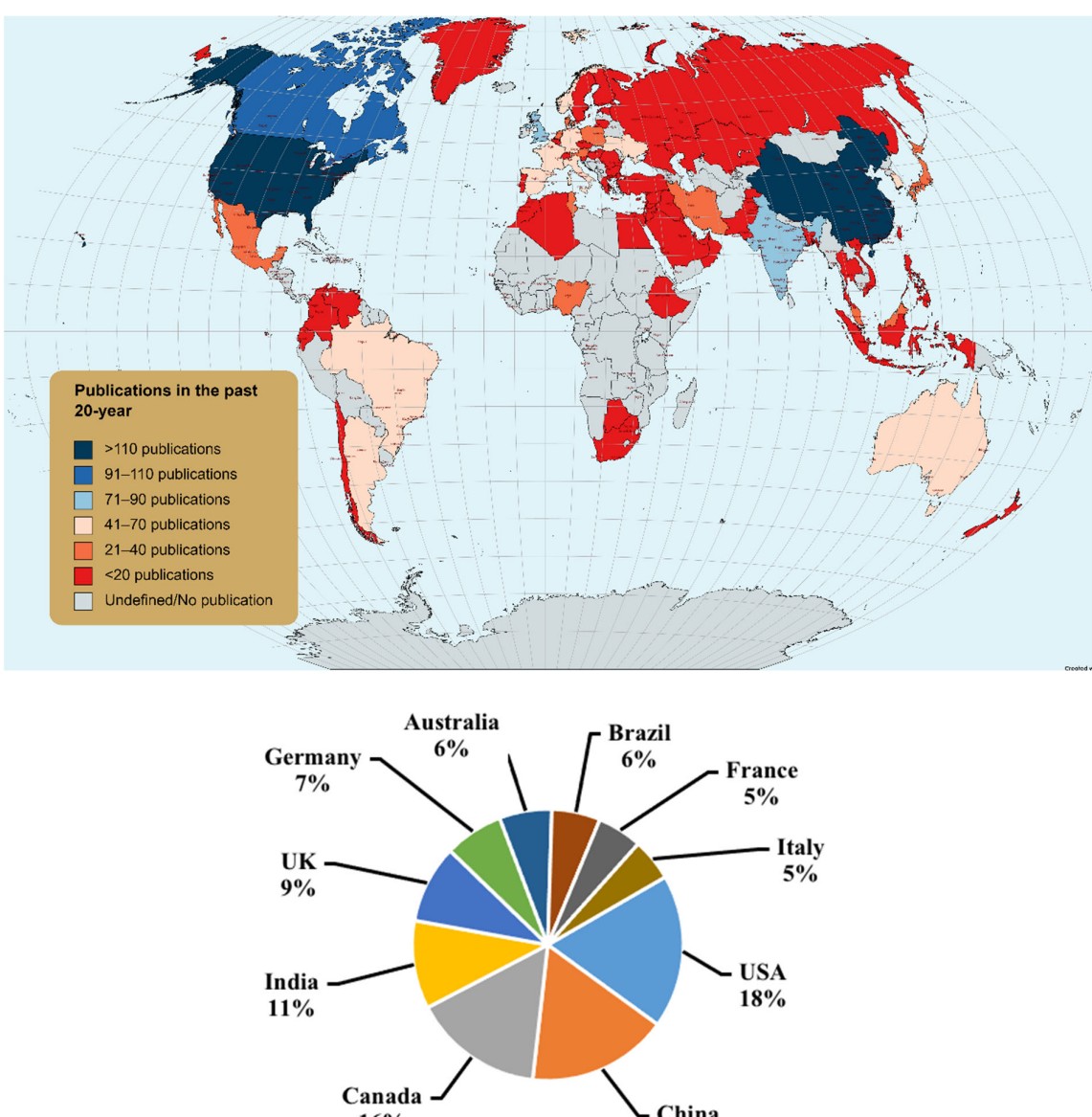

**Figure 4.** Global publication outputs (**top**); percentage of articles published by top 10 countries (**bottom**) pertinent to Antarctic hydrocarbon bioremediation. Distribution map was generated through online tool [16,19].

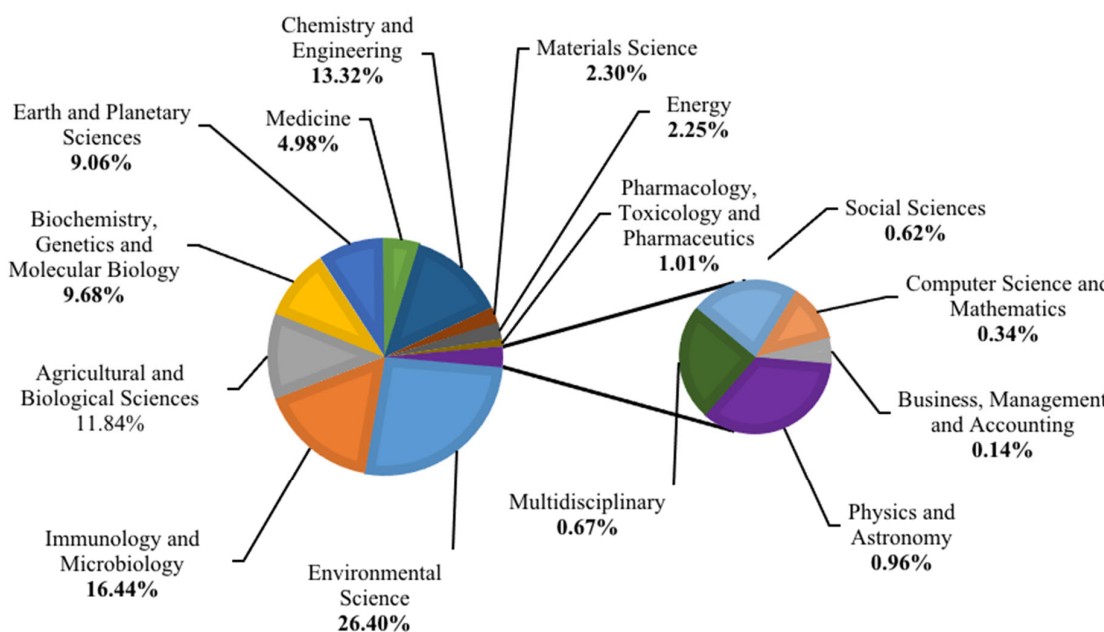

**Figure 5.** Percentage distribution of studies related to biodegradation of hydrocarbons in Antarctica by subject area.

*3.3. Keywords Co-Occurrences Analysis for Publications in 20-Year Spans and Future Research Direction*

The keywords represent the research focus, meant to convey the gist of a whole article oftentimes listed in multiple numbers. Various bibliometric analysis tools (e.g., VOSviewer, Gephi, Pajek and CiteSpace) usually have the built-in function for keywords co-occurrences analysis, which measures the number of terms repeated within the chosen theme [20]. In this review, the search parameters were refined to exclusively narrow down the results to research articles relevant to "diesel bioremediation" and "Antarctic bacteria" circa 2000–2020, with a threshold value of at least three keyword occurrences. From the total selected of 751 articles, 418 keywords appeared more than twice, yet, only 300 items were linked together, requiring manual scrubbing through the list to eliminate irrelevant subjects.

The visual map generated creates an overview of multiple disciplines involved in Antarctic-relevant studies on hydrocarbon contaminants, which then can be clustered further into a larger area of studies (Figure 6). The authors have identified three study domains with themes entailing (i) the strategies for hydrocarbon remediation in Antarctica, (ii) the anthropogenic impacts and problems management, and (iii) the sustainability studies for fuel replacements.

The size of circles in the diagram represents the frequency of that particular keyword for a fixed number of times in the constructed database, while the line thickness and closeness between nodes is relative to the degree of connection between disciplines, often deduced from simultaneous co-appearance of terms within the same publication [5]. The keyword "Antarctica" appeared as the central theme with the highest total link strength (TLS = 305), while other significant nodes located close to it were "diesel" and "biodegradation" with TLS = 135 and TLS = 105, respectively. These findings impart a notion that diesel fuel is likely the key contributor to the current Antarctic's hydrocarbon pollution index and suggest that it is often paired to those studies of biodegradation as the preferred practice in tackling the issues over other conventional approaches. Upon breaking down the quantitative analysis, the conclusion drawn from Figure 6 is further supported by data presented from Antarctic Treaty Inspection Programme Report [21] which conducted surveys on approximate annual fuel consumptions (diesel only) from various stations during the 2014/2015 season.

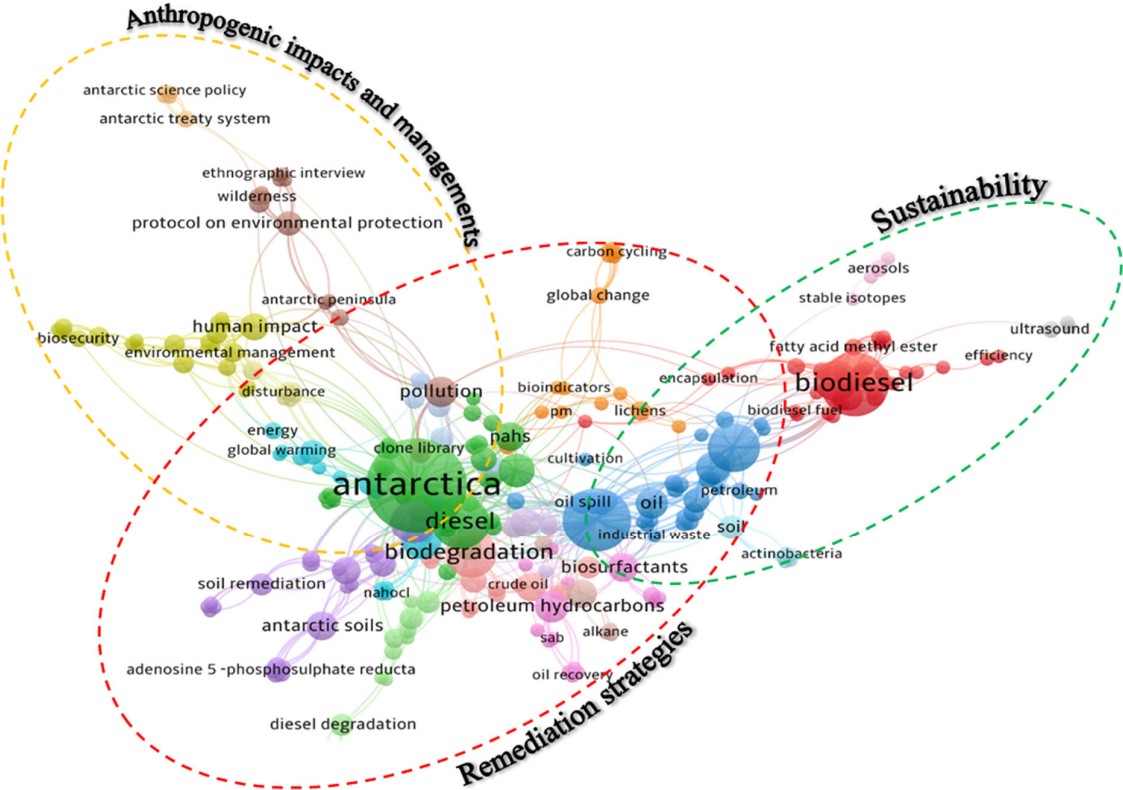

**Figure 6.** Research topics from keywords co-occurrences. Dashed lines highlighted the areas of knowledge proposed by the authors.

Although some stations are now opting for alternative sources such as solar power, wind turbine and tidal energy, diesel fuel remains as the principal source of energy hitherto. Referring to the overlay diagram in Figure 7 that shows the emergence of research keywords relative to a specified timeframe, it can also be observed that Antarctic scientists are actively conducting studies on biodiesel production from general wastes (e.g., food refuse, used cooking oil and algae) since 2015 until the present time (indicated by teal to yellow shading) linked to hydrocarbon pollution (LTS = 72). This proposed that the research direction is not in favour of conventional renewable energy sources, probably due to severe limitations.

On the other hand, among the most common bioremediation−related keywords were biostimulation, bioaugmentation, biopile and biocomposting, which demonstrated multiple available approaches to remediate hydrocarbon pollutants at low-temperature have been trialled and for the most of the part stayed relevant after 20 years. The authors' topic of interest, which is the optimisation of hydrocarbons-degrading bacteria in terms of author keywords, was seen to be directly related (TLS = 24, six occurrences) to similar studies revolving around "Antarctica", "response-surface methodology", "growth kinetics", "bioremediation" and "diesel fuel", appearing relatively as a small niche yet impactful enough to be shown on the network map. Besides the two aforementioned knowledge areas, the studies on the anthropogenic impacts and managements were seen to increase in numbers even though they are located further from the centre. Table 1 shows the top 15 keywords in which some of the terms are related to the policy-making areas. Human impacts are largely recognised, which relates back to Figures 3 and 7 that linked the increase in civilian and military activities to the current pollutions, thus raising questions whether stricter regulations are ever needed to secure the future of Antarctica.

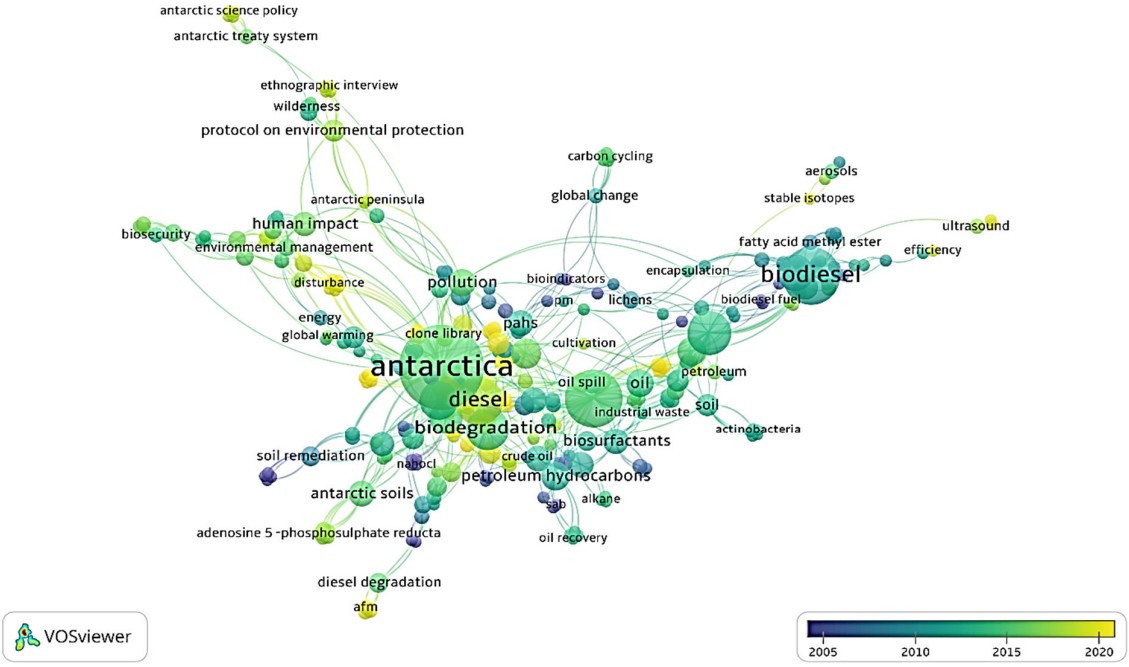

**Figure 7.** Network overlay diagram showing emergence of research topics in relative to year of publication.

**Table 1.** The occurrences and total link strength pertaining to bioremediation of hydrocarbons in Antarctica.

| Keyword | Occurences | Total Link Strength |
|---|---|---|
| Antarctica | 75 | 240 |
| Bioremediation | 35 | 122 |
| Diesel | 26 | 97 |
| Bacteria | 11 | 39 |
| Human impacts | 7 | 35 |
| Policy | 4 | 30 |
| Antarctic soils | 8 | 28 |
| Heavy metals | 7 | 28 |
| Pollution | 7 | 27 |
| Biostimulation | 7 | 26 |
| Antarctic Treaty | 4 | 24 |
| Conservation | 4 | 24 |
| Marine | 6 | 24 |
| Response surface methodology | 6 | 24 |
| Biosurfactants | 6 | 22 |

### 3.4. Studies of Hydrocarbonoclastic Microbes for Low-Temperature Bioremediation

The detrimental effects of many diesel constituents are largely associated to the polycyclic aromatic hydrocarbons (PAHs) derivatives and the heavy-chained hydrocarbons, requiring effective and rapid removal upon spillage to prevent further damage to the surrounding environment [22]. In recent decades, bacterial bioremediation has garnered worldwide attention for its state-of-the-art application, which exploits specific biochemical pathways of certain organisms to convert hydrocarbons of varying molecular masses into structurally simpler and less harmful metabolic products, essentially the carbon dioxide, water and bioenergy while simultaneously restores the condition of a polluted site [23,24]. Presently, large portions of the commonly ventured bioremediation strategies are the bioaugmentation, biostimulation, phytoremediation and bioreactor-based remediation, in which some were employed due to their relatively cost-effective and less destructive approach, although at the expense of longer time for noticeable effects and greatly inconsistent success rates from multiple factors interaction [25–27].

For microbial remediation of hydrocarbons, the biogeochemical factors (pH, temperature, salinity, C:N:P ratio and oxygen content) govern the substrate bioavailability towards degradation, which also correlates directly to the degradation rate [28,29]. Apart from the aforementioned factors, the physicochemical nature of the diesel itself (concentration of varied fuel components, substitution groups, viscosity and solubility) may serve as additional elements affecting the total degradability of petroleum hydrocarbons [30]. In terms of general biodegradability, the susceptibility of petroleum hydrocarbons decreases from paraffins > branched alkanes > olefins > monocyclic aromatic hydrocarbons (MAHs) > naphthenes > polycyclic aromatic hydrocarbons (PAHs) [31]. On top of the heightened resistance towards assimilation as the molecular structure gets more complex, the extreme cold of Antarctica also turns them into persistent environmental pollutants by further limiting water solubility and susceptibility to nucleophilic attacks, hence affecting their overall bioavailability for bacterial degradation [32]. Throughout the past decades, studies have identified and characterised a range of naturally-occurring bacterial genera with decent potential for oxidative degradation of thermogenic hydrocarbons including Rhodococcus sp., Burkholderia sp., Acinetobacter sp. and *Pseudomonas* sp. [22,25,33,34], while fungi and yeast from Aspergillus, Candida, Penicillium and Rhodotorula genera have also been reported widely as potential candidates in hydrocarbon degradations [31,35,36].

In reality, nature itself will act upon the source of pollution until equilibrium is achieved without human intervention; this mechanism is called natural attenuation, which refers to the degradation of in situ contaminants by autochthonous microbes coupled with favourable biogeochemical factors [37,38]. However, when the severity of pollution exceeds the natural capability to correct the imbalances (often deduced from the declining in population size of living organisms in the affected site) and the ambient conditions that are too harsh to support mass bacterial propagation (especially Antarctica), this clearly demands for human intervention. As an extension to the latter, the application of biostimulation may improve the biodegradation efficiency of autochthonous organisms by controlled supplementation of limiting nutrients such as chemical fertilisers [C:N:P], agronomic wastes and/or animal manures to simulate the ideal conditions for microbial propagation [39,40]. Additionally, the use of naturally-produced biodegradable surfactants accelerates the dispersion of hydrocarbon molecules, hence increasing the substrates bioavailability by a substantial amount, especially towards that of petroleum-originated, highly hydrophobic contaminants [41,42]. Factually, previous studies pertaining to petroleum degradation have documented an average increment of about 25% to 50% of degradation efficiency in comparison to the natural attenuation alone [43,44]. While principally similar to biostimulation, the bioaugmentation approach enhances hydrocarbons biodegradation through the introduction of natural or genetically modified exogenous microbes with specific catabolic activities at the polluted site [45]. According to Tyagi et al. [46], bioaugmentation with bacterial consortium is highly recommended for its synergistic effects compared to that of monocultures which have catabolic capacity limited to only certain petroleum components. Henceforth, many studies on thermogenic hydrocarbons biodegradation are now shifting their focus to using bacterial communities and single-celled yeast/fungi (Table 2) [42,47], while phytoremediation in Antarctica is seldomly studied, simply due to the lack of vegetation able to degrade diesel [25,48].

Table 2. Examples of pure and mixed hydrocarbonoclastic microorganisms isolated from Antarctica during the last 20 years.

| Microorganism(s) | Substrate(s) Degraded | Initial Concentration | Maximum Efficiency (%) | Location of Study/Source of Sample | Method of Study | Reference |
|---|---|---|---|---|---|---|
| **Bacteria** | | | | | | |
| *Pseudomonas* Ant 5 and *Sphingomonas* Ant 17 | BTEX, naphthalene, JP-8 fuel | Various | NA | Scott Base (Ross Island, Antarctica) | Cell culture | [22] |
| *Arthrobacter* sp. E28 and *Rhodococcus* sp. E60 | Diesel | 1% v v$^{-1}$ | 86% E28 and 89.2% E60, 160 days | Terra Nova Bay (Ross Sea, Antarctica) | Cell culture | [49] |
| Soil microbial community | n-Alkane, PAHs, toluene, Diesel, hexadecane | 5 μL ml$^{-1}$ | NA | Scott Base (Ross Island, Antarctica) | RFLP | [50] |
| *Halomonas* sp. ANT 3b | | 2% v v$^{-1}$ | NA | Terra Nova Bay station (Ross Sea, Antarctica) | Cell culture | [51] |
| *Pseudomonas* sp. ST41 | Polar Blend marine gasoil | 1% w dw$^{-1}$ | NA | Signy Island (South Orkney Islands, Antarctica) | TGGE, bioaugmentation and biostimulation | [52] |
| Soil microbial community | Special Antarctic Blend (SAB) | 10,000–47,000 ± 630 mg kg$^{-1}$ | NA | Casey Station (Bailey Peninsula, Antarctica) | DGGE | [53] |
| *Rhodococcus* sp. and *Alcaligenes* sp. co-culture | Diesel, n-alkane | 2% v v$^{-1}$ | NA | Terra Nova Bay (Ross Sea, Antarctica) | ARDRA and RAPD | [54] |
| Soil microbial community | Phenanthrene | 0.14–1.47 ng g$^{-1}$ dw | 47.93%, 7 days | Livingstone Island (South Shetlands Islands, Antarctica) | Cell culture | [55] |
| *Planococcus* sp. NJ41 and *Shewanella* sp. NJ49 | Diesel, naphthalene, hexadecane | 50 mg L$^{-1}$ | NA | Antarctic Ocean | Cell culture | [56] |
| Soil microbial community | Diesel | 2180 mg kg$^{-1}$ dw | 75.79% BP and 49.54% BS, 50 days | Carlini station (South Shetlands Islands, Antarctica) | RSM, Biopile (BP) and biostimulation (BS) | [57] |
| Soil and sediment microbial communities | Diesel | 200–1000 μg g$^{-1}$ dw | NA | Carlini station (South Shetlands Islands, Antarctica) | DGGE | [58] |
| *Sphingobium xenophagum* D43FB | Phenanthrene | 2000 ppm | 95%, 5 days | King George Island, South Shetland Islands, Antarctica | Cell culture | [59] |
| Marine sediment microbial communities | Crude oil and diesel | 1.5% v v$^{-1}$ | NA | Livingston Island (Byers Peninsula, Antarctica) | T-RFLP and DGGE | [60] |
| *Arthrobacter* sp. AQ5-05 | Diesel | 3% v v$^{-1}$ | 56.32%, 10 days | King George Island (South Shetland Islands, Antarctica) | RSM, cell culture | [34] |
| *Rhodococcus* sp. AQ5-07 | Diesel | 1% v v$^{-1}$ | 90.39%, 7 days | King George Island (South Shetland Islands, Antarctica) | RSM, cell culture | [33] |
| Soil microbial communities | Antarctic Gasoil, AGO | 7620 ± 680 mg kg$^{-1}$ dw | 87 ± 13%, 1 year | Carlini station (South Shetlands, Antarctica) | DGGE | [61] |
| **Fungi/Yeast** | | | | | | |
| Various yeast strains | Hexadecane dodecane, Undecane | 1 g L$^{-1}$ | 39.9% and 8 days | Continental glacier, Antarctica | Cell culture | [62] |
| *Candida antarctica* T-34 | | | 83.9 ± 1.2 % | Ohridski Base, Livingston Island, Antarctica | Cell culture | [63] |
| *Exophiala* sp. and *Pseudeurotium bakeri* | Special Antarctic Blend (SAB) | 50–20,000 mg kg$^{-1}$ | NA | Australian Research station (Macquarie Island, Antarctica) | RFLP | [64] |
| *Pichia caribbica* | n-alkanes and diesel fuel | 1 g L$^{-1}$ | NA | Carlini Station (South Shetlands Islands, Antarctica) | Cell culture | [65] |

### 3.5. Genes Families Conferring to Hydrocarbons Catabolisms and Cold-Adaptation Features in Antarctic Microbes

Hydrocarbons, especially the complex molecules contain many C-H linkages and are less susceptible to microbial degradation while in their inert form [31]. In order to degrade thermogenic hydrocarbons, they need to undergo cascades of complex catabolism processes that may proceed aerobically or in absence of oxygen as the initial electron donor. The preferred degradation mode, being oxygen-dependent pathways, allow for complete transformation of hydrocarbon molecules into carbon dioxide, water and cellular energy, while some extreme chemoautotrophs may resort to anaerobic pathways to overcome oxygen limitation [66]. Aliphatic and branched hydrocarbon degradation is initiated by upstream enzymes including monooxygenases which activate the alkane through the oxidation of monoterminal, diterminal or subterminal hydrogen atom to form primary or secondary n-alkanol [67] In addition, another group of enzymes closely related to the cytochrome P450 system called hydroxylases similarly catalyse the hydroxylation of alkane to that of monooxygenases [68]. Beyond the first step, downstream pathways of activated-alkanol degradation vary among organisms, depending on its gene constituents. Generally, the complete degradation of hydrocarbons through the monoterminal pathway follows the succession from n-alkane to n-alkanol by hydroxylases/monooxygenases which then continues to be oxidised by membrane-bound alcohol dehydrogenase to produce n-alkanal before it is further transformed into fatty acid where the product is likely to enter the fatty acid β-oxidation to generate acetyl Co-A at the expense of producing two carbon atom molecules in each cycle [69]. However, a diterminal pathway occurs through the oxidation of both terminals via ω-oxidation to form dicarboxylic acid before it can enter β-oxidation. Meanwhile, the subterminal oxidation pathway will see the oxidation of secondary alcohol into ketone and then ester, before being further hydrolysed into fatty acid and alcohol [68].

Besides alkanes, aromatic hydrocarbons are also major constituents of modern-day diesel fuels, which they are even harder to be metabolised as compared to alkanes. The presence of C=C double linkages hinders nucleophilic attack, thus causing the molecule to be more persistent in nature. The bacterial degradation of te aromatic ring proceeds through the initial oxidation of the double bond to form transdiol, followed by ring cleavage to obtain a dicarboxylic acid [70,71]. The cleaving of benzene ring can occur in two distinguished positions; the ortho- and metapathways, resulting in the generation of catechols and procathecuates, which will be broken down and enter the TCA cycle as central intermediates [69]. The study by Cao et al. [72], determined that low benzoate concentration induces only the orthocleavage, while sufficient concentration will activate both pathways. However, the level that induces both pathways differs between species. The gene catR and low concentration cis, cis-muconate both repress the operon for the orthopathway. In more complex molecules such as PAHs, the degradation follows the same steps. However, it occurs through cleaving one ring at a time. Despite using varying degradation pathways which correspond to different genes present in bacterial species (Table 3), the hydroxylases/monooxygenases remain the pivotal entities in the whole hydrocarbon degradation metabolism [73].

**Table 3.** List of genes involved in aromatic hydrocarbon degradation sorted in reaction orders and the respective translated enzymes [74–76].

| Gene Name | EC Number | Enzyme Name |
|---|---|---|
| | **Cyclohexane (via β-oxidation)** | |
| bmoX | - | cyclohexane hydroxylase |
| chnA | 1.1.1.245 | cyclohexanol dehydrogenase |
| chnB | 1.14.13.22 | cyclohexanone-NADPH monooxygenase |
| chnC | 3.1.1.- | caprolactone hydrolase |
| chnD | 1.1.1.258 | hydroxyhexanoate-NAD$^+$ dehydrogenase |
| chnE | 1.2.1.63 | oxohexanoate dehydrogenase |
| | **Ethylbenzene (Via procathecuate)** | |
| ebdA | | ethylbenzene dehydrogenase-α |
| ebdB | 1.17.99.2 | ethylbenzene dehydrogenase-β |
| ebdC | | ethylbenzene dehydrogenase-γ |
| ped | 1.1.1.311 | (S)-1-phenylethanol dehydrogenase |
| apcA | | acetophenone carboxylase |
| apcB | | acetophenone carboxylase |
| apcC | 6.4.1.8 | acetophenone carboxylase |
| apcD | | acetophenone carboxylase |
| apcE | | acetophenone carboxylase |
| bal | - | benzoylacetate-CoA ligase |
| fadA | - | 3-keto-acyl-CoA-thiolase |
| | **Shikimate/quinate degradation (via procatechuate)** | |
| quiA | 1.1.5.8 | shikimate dehydrogenase |
| quiA | | quinate dehydrogenase |
| quiB | 4.2.1.10 | dehydroquinate dehydratase |
| quiC | 4.2.1.118 | dehydroshikimate dehydratase |
| | **Hydroxymandelate degradation (via procatechuate)** | |
| mdlA | 5.1.2.2 | (S)-4-hydroxymandelate racemase |
| mdlA | | mandelate racemase |
| mdlB | 1.1.5.- | (S)-mandelate dehydrogenase |
| mdlB | | (S)-2-hydroxy-2-(4-hydroxyphenyl)acetate:acceptor 2-oxidoreductase |
| mdlC | 4.1.1.7 | *p*-hydroxybenzoylformate carboxy-lyase |
| mdlD | 1.2.1.96 | NADP+-4-hydroxybenzaldehyde dehydrogenase |
| pobA | 1.14.13.2 | *p*-hydroxybenzoate hydroxylase |
| pobA | | *p*-hydroxybenzoate hydroxylase |
| | **Procatechuate degradation (via 2-hydroxypenta-2,4-dienoate)** | |
| praA | 1.13.11.- | protocatechuate 2,3-dioxygenase |
| praH | - | 5-carboxy-2-hydroxymuconate-6-semialdehyde decarboxylase |
| praB | 1.2.1.85 | 2-hydroxymuconate-6-semialdehyde dehydrogenase |
| xylG | | 2-hydroxymuconate semialdehyde dehydrogenase |
| praC | 5.3.2.6 | 4-oxalocrotonate tautomerase |
| xylH | | 2-hydroxymuconate tautomerase |
| praD | 4.1.1.77 | 2-oxo-3-hexenedioate decarboxylase |
| xylI | | 2-oxo-3-hexenedioate decarboxylase |
| | **Toluene degradation (via 2-hydroxypenta-2,4-dienoate degradation)** | |
| todC2 | | toluene 1,2-dioxygenase |
| todC1 | 1.14.12.11 | toluene 1,2-dioxygenase |
| todA | | toluene 1,2-dioxygenase |
| todB | | toluene 1,2-dioxygenase |
| todD | 1.3.1.19 | cis-toluene dihydrodiol dehydrogenase |
| todE | 1.13.11.2 | 3-methylcatechol 2,3-dioxygenase |
| todF | 3.7.1.25 | 2-hydroxy-6-oxohepta-2,4-dienoate hydrolase |
| | **Mandelate degradation (via catechol)** | |
| mdlA | 5.1.2.2 | mandelate racemase |
| mdlB | 1.1.99.31 | (S)-2-hydroxy-2-phenylacetate:acceptor 2-oxidoreductase |
| mdlC | 4.1.1.7 | benzoylformate carboxy-lyase |
| mdlD | 1.2.1.7 | NADP+-benzaldehyde dehydrogenase |

**Table 3.** *Cont.*

| Gene Name | EC Number | Enzyme Name |
|---|---|---|
| ntnC | | benzaldehyde dehydrogenase |
| xylC | 1.2.1.28 | benzaldehyde dehydrogenase |
| xylX | | benzoate 1,2-dioxygenase |
| xylY | 1.14.12.10 | benzoate 1,2-dioxygenase |
| xylZ | | benzoate 1,2-dioxygenase |
| xylL | 1.3.1.25 | 1,2-dihyroxy-3,5-cyclohexadiene-1-carboxylate dehydrogenase |
| **Naphthalene degradation (via catechol)** | | |
| nahAd | | naphthalene 1,2-dioxygenase |
| nahAc | | naphthalene 1,2-dioxygenase |
| nahAb | | naphthalene 1,2-dioxygenase |
| nahAa | | naphthalene 1,2-dioxygenase |
| ndoA | 1.14.12.12 | naphthalene 1,2-dioxygenase |
| ndoC | | naphthalene 1,2-dioxygenase |
| ndoB | | naphthalene 1,2-dioxygenase |
| ndoR | | naphthalene 1,2-dioxygenase |
| nahB | 1.3.1.29 | cis-1,2-dihydro-1,2-dihydroxynaphthalene-1, 2-dehydrogenase |
| nahC | 1.3.11.56 | 1,2-dihydroxynaphthalene dioxygenase |
| nahD | 5.99.1.4 | 2-hydroxychromene-2-carboxylate isomerase |
| nahE | 4.1.2.45 | trans-o-hydroxybenzylidenepyruvate hydratase-aldolase |
| nahF | | salicylaldehyde dehydrogenase |
| alkH | 1.2.1.65 | aldehyde dehydrogenase |
| **Salicylate degradation (via catechol)** | | |
| salA | | salicylate 1-hydroxylase |
| nahW | | salicylate hydroxylase |
| nahG | | salicylate hydroxylase |
| **Anthranilate degradation (via catechol)** | | |
| kyn | 3.7.1.3 | kynureninase |
| antB | | anthranilate dioxygenase |
| antA | 1.14.12.1 | anthranilate dioxygenase |
| antC | | anthranilate dioxygenase |
| **Catechol degradation (via 2-hydroxypenta-2,4-dienoate)** | | |
| xylE | 1.13.11.2 | catechol 2,3-dioxygenase |
| xylF | 3.7.1.9 | 2-hydroxymuconic semialdehyde hydrolase |
| **2-Hydroxypenta-2,4-dienoate degradation (via acetyl-CoA)** | | |
| todG | | 2-oxopent-4-enoate hydratase |
| cmtF | 4.2.1.80 | 2-oxopent-4-enoate hydratase |
| xylJ | | 2-oxopent-4-enoate hydratase |
| cmtG | | 4-hydroxy-2-oxovalerate aldolase |
| todH | 4.1.3.39 | 4-hydroxy-2-oxovalerate aldolase |
| xylK | | 4-hydroxy-2-oxovalerate aldolase |
| todI | | acylating aldehyde dehydrogenase |
| cmtH | 1.2.1.0 | acetaldehyde dehydrogenase |

Instances of same gene in different degradation pathways indicate multifunctional enzymes. Same enzyme name corresponding to the EC number indicates subunits-complex.

Apart from the genes responsible for hydrocarbons degradation, physiological adaptations are also imperative for the survival of bacteria at frigid Antarctic temperatures. The subzero environment imposes cold shock, which reduces both membrane fluidity and enzymatic activities, hinders transcription and translation of genomic materials and limits the free water molecules required to maintain cellular activities [77]. However, the notion that bacterial activities are in complete cessation at subfreezing temperatures is not entirely true. Notable acclimatisation in psychrotolerants include the expression of cold-shock and antifreeze proteins, membrane modifications which increase the content of membrane-bound unsaturated and branched fatty acids and carotenoids, as well as the production of cold-stable isozymes with varying changes in amino acid abundance [78,79]. These adaptations can be apparent at genome level (rather than at the level of individual genes) due to differences between organisms resulting from genetic drift or the specific

environment. Many comparative studies pertaining to cold-adaptation modifications have implemented the approaches of comparing the cold/hot ratio of amino acids affiliated to membrane flexibility (i.e., proline, lysine, serine and glycine), measuring the expression levels of cold-shock, heat-shock and antifreeze proteins, as well as comparing the whole genetic makeup of the organisms in phylogeny [80]. Table 4 illustrates the typical genes investigated in cold-adapted organisms.

**Table 4.** Some of the known and predicted genes related to cellular growth and cold adaptability of bacteria [78,81–85].

| **Heat-Shock (Hsp) and Cold-Shock (Csp) Proteins** |
| :---: |
| Molecular chaperone GrpE (heat shock protein) |
| Molecular chaperone DnaK (HSP70) |
| Molecular chaperone IbpA, HSP20 family |
| Ribosomal 50S subunit-recycling heat shock protein |
| Chaperonin GroEL (HSP60 family) |
| Co-chaperonin GroES (HSP10) |
| Cold shock protein, CspA family |
| Universal stress protein, UspA family |
| **Membrane and peptidoglycan modification** |
| 3-oxoacyl-[acyl-carrier-protein] reductase |
| Glycosyltransferase involved in cell wall biosynthesis |
| Fatty-acid desaturase |
| D-alanyl-D-alanine carboxypeptidase |
| **Polysaccharide envelope** |
| Capsule polysaccharide export protein |
| Capsular polysaccharide biosynthesis protein, EpsC |
| Exopolysaccharide biosynthesis protein |
| **Carotenoid biosynthesis** |
| Phytoene/squalene synthetase |
| Phytoene dehydrogenase-related protein |
| **Osmotic and oxidative stress response** |
| ABC proline/glycine betaine transport, ATPase and permease |
| Osmoprotectant binding protein |
| Choline dehydrogenase or related flavoprotein |
| Trehalose-6-phosphate synthase |
| Na+/proline symporter |
| Na+/H+ antiporters |
| Catalase |
| Glutathione peroxidase |
| Glyoxylase or related hydrolase, $\beta$-lactamase superfamily II |
| **Translation and transcription factors** |
| Translation elongation factor EF-Tu/G, GTPase |
| Translation initiation factor IF-2/3, GTPase |
| Transcription antitermination factor NusA |
| Transcription termination factor NusB |
| Superfamily II DNA or RNA helicase, SNF2 |

*3.6. Common Community Identification Techniques Applied in Bioremediation*

The fact that gene compositions vary greatly between organisms suggest that microbially-mediated hydrocarbon removal should be attempted using a consortium approach as it will provide an enormous pool of hydrocarbon-degrading genes in comparison to a single organism. In this sense, conventional genomics or relatively advanced metagenomics approaches can be used to screen for intrinsic hydrocarbonclastic capability as well as the metabolic limit of microbial community [86]. At its core, metagenomics is an interdisciplinary approach encompassing well established multiomics, including but not limited to genomics, systems biology and bioinformatics. While metagenomics is relatively a young branch of science, the development of various culture-independent identification methods through this principle is very fast-paced in parallel to the advances of modern

technologies (i.e., machine, algorithm and chemical) [87]. Although several techniques have been invented hitherto, the extraction of biomolecules may still be compromised in many ways including the ubiquitous humic acids content in soil and the presence of interfering molecules or metal ions from contamination. Therefore, the selection of method is reflective of what the objective of the study is, as well as taking into account the possible sources of interference in the soils.

### 3.6.1. Phospholipid Fatty Acid Analysis (PLFA) and Fatty Acyl Methyl Ester (FAME) Analysis

The omnipresence of membrane-bounded fatty acids (i.e., phospholipids, lipopolysaccharides etc.) and signature metabolized lipid products can be an indicator of a particular taxonomic group within a community [88]. Thus, the unique lipid compositions between organisms form the basis of genomics such as PLFA and FAME analyses. FAMEs are products of alkali digestion (saponification) of bacterial membrane phospholipids bilayer followed by methylation/acylation. Derivatization is chemically imparting minor modification to the fatty acids' properties through acidification or alkalization for better resolution in gas chromatography analysis [89]. The resulting chromatograph profiles can be cross-referenced against available FAMEs databases to identify and classify the fatty acids and their corresponding microbial signatures by multivariate statistical analyses [90]. These patterns are stable phenotypic expressions rather than genotypic, which makes them highly conserved due to their pivotal role in membrane structure and function. The immediate decomposition of phospholipids upon cell death means lipid-based approaches are able to discriminate viable and nonviable microbial groups together with their relative abundance, physiological status and overall community compositions [91]. General criteria in both PLFA and FAME analyses for bacterial identification include i) variation in chain lengths or branching, ii) degree of saturation/hydroxylation, and iii) presence of distinct functional groups (cyclopropane, methyl, esters) [92]. Comparatively, since the extraction step in FAME analysis is performed directly on soil samples, therefore the lipid profile is wider as it does not dictate the sources of FAs, while PLFA approach involves additional fractionation of lipid components to account only the phospholipids. Further optimizations at multiple steps during the extraction are expected to reduce the FAs yields from nonmicrobial sources to below 10% contamination level, significantly increasing the sensitivity and accuracy of analysis [93]. These identification approaches have been described as equally efficient, rapid and reproducible for both clinical and environmental studies involving heavy metal pollution, soils chemical augmentation and perturbation.

### 3.6.2. Non-PCR-Based Nucleic Acids Analyses

Traditionally, the stronger stacking interactions between guanine and cytosine (GC) contents have been used to characterize microorganisms due to the triple hydrogen bonding that confers structural stability to nucleic acids [94]. Based on this knowledge, it is expected that GC richness will differ between organisms and may become more apparent as they evolve further apart phylogenetically [95]. Characterization through this approach alone is relatively outdated by today's standards, simply due to its low resolving depth especially when involving organisms with similar GC range. Nevertheless, it is very common to see GC contents analysis being incorporated to more sophisticated techniques as supplementary data.

Other than that, the measure of DNA strand hybridization rate may also loosely estimate the abundance of microbial diversity within a community. As the complexity of bacterial compositions increases, it affects the amount of different DNA sequences extracted. When kinetically measured at specific conditions, a sample with more DNA sequences take longer time to anneal, thus serving as diversity indicator. The index is expressed as the half association value C0t1/2, controlled by the concentration of DNA product and time of incubation [96]. In the improvised method using the same principle, a competitive DNA hybridization method, also coined as reverse sample genome probing (RSGP), can be used to identify targeted groups of organisms within a sample [97]. Through

this approach, a biomarker sequence unique to the organisms of interest is used as a probe to competitively anneal to the homologous microbial DNA, in which the rate of biomarker association will tell the partial composition of the sample [98]. Moreover, according to Shanks et al. [98], RSGP is very useful for targeted microbial studies, in which identification of the whole community is not the goal, while still maintaining the resolution power needed for bacterial identification.

### 3.6.3. PCR-Based Nucleic Acids Analyses

In PCR-dependent bacterial identifications, the highly conserved, short genomic sequence of 16S or 18S rDNA region is amplified by polymerase chain reaction and analyzed accordingly through in vitro then in silico, based on the selected approach. Through restriction fragment length polymorphism (RFLP) analysis, the PCR-amplified conserved region is digested with a defined set of endonucleases to produce variable-length fragments termed 'DNA fingerprints' that differs between organisms, which can be distinctly visualized through electrophoretic separation using capillary-based or gel-based approaches [99]. Later improvisation, the terminal restriction fragment length polymorphism (T-RFLP) also includes terminal fluorescein-tagged primers to simplify the pattern interpretation, wherein each florescent band represents a single operational taxonomic unit (OTU) that measures the homogeneity and community richness within or between samples [100]. The measurements can be expressed differently to represent certain aspects of richness; such as the Pielou index (evenness), Shannon–Weiner or Simpson indices (diversity), Sorensen or Jaccard indices (similarity), and Dice index (dynamic shift). DNA polymorphism is a considerable genetic variance between individuals in a population with little to no apparent effects on gene functionality. Often mistaken for mutations which are deleterious at gene level, polymorphisms do not interfere with either the gene or protein structure and can be passed down through generations from the point of formation [101]. There are several classes of DNA polymorphisms, the most common being single-nucleotide polymorphisms (SNPs) and the variable number tandem repeats (VNTRs/microsatellites) [102].

Apart from using restriction enzymes, polymorphic profiles can also be constructed via denaturing gradient gel electrophoresis (DGGE) using chemical denaturants (e.g., urea, formamide). Here, the application of 40–50 bp, GC rich primer is very crucial to prevent the loss of PCR products due to complete strand dissociation at higher denaturant concentrations. Despite the differences in the amplified DNA region and length, a number of comparative studies have reached to a consensus that i) patterns obtained are very similar across different approaches and ii) DNA fingerprint clustering strongly correlates to soil physicochemical properties [103,104]. Notable drawbacks of using these methods include the limited number of separation bands (<100 per gel slab), potentially leading to underestimation of true microbial diversity, and the overlapping of OTUs from different bacterial species (OTU homoplasy) [90].

In high-resolution melting curve analysis (HRM), the principle of gradient thermal degradation is coupled to reverse transcriptase quantitative PCR (RT-qPCR) for high-accuracy population and gene expression studies. The process begins with amplification of 16S rDNA region, while the thermocycler concurrently measures the product concentration in real-time at the end of each cycle [105]. During PCR synthesis, amplicons are intercalated with fluorescent dye (e.g., SYBR green, SYTO9 or Chomofy) to form tagged-dsDNA and allowed to completely rehybridize before slow gradient melting, typically from 50–95 °C, is applied with 0.1 °C increment between readings [106]. As the melting temperature, Tm, of the amplicon is reached, the double-stranded DNA unwinds, and thus gradually loses measurable fluorescence in the process. Melt curves generated after HRM analysis are plotted in real-time with temperature on the X axis against fluorescence intensity on the Y axis, and are very useful in elucidating the characteristics of DNA being tested whether it is homozygous wild-type, homozygous mutant or heterozygous wild-type and mutant [105].

### 3.6.4. Sequencing-Based Nucleic Acid Analyses

Principally, the metagenomics approach sequenced and analyzed the entirety of genetic materials in the microbial community in the manner of whole-genome sequencing (WGS), as opposed to partial identification using targeted genes sequencing and DNA fingerprinting discussed previously. The ability to generate millions of reads in shorter time and number of resources, without excessive trade-off in precision has totally eclipsed earlier methods of microbial identification.

In one of the metagenomic approaches, shotgun sequencing involves the shearing of gDNA into random fragments of 1–2 kbp in length which are cloned into vectors (plasmid, cosmid, BAC and YAC) to generate genomic libraries [107]. While the partially sequenced fragments themselves are not sufficient to identify microbial compositions, bioinformatical de novo assemblies (Figure 8) based on sequences overlaps 'glue' the pieces together into long, recognizable sequences called contigs, usually separated by gaps of unknown sequences. The various causes of gap formation, such as unclonable regions, presence of repetitive sequences and DNA secondary structures, are inevitable, thus lower contigs number signifies ideal sequencing protocol [108]. Presumably, the blank space can be filled through (i) hybridization of cloned sequence/probe that can bind to both ends of contigs or (ii) running a PCR on whole gDNA with pairs of primers corresponding to the contigs ends, although the latter can be resource-consuming in order to identify the correct pairs [109].

For a successful, complete identification that represents the whole microbial community, a minimum clone-library size in the order of between six to eight times the original genomic concentration is highly recommended. Although shotgun metagenomic sequencing does not involve the biased amplification of 16S rRNA regions, the DNA extraction and sequencing protocols employed in metagenomics greatly affects the relative organism abundances at the end of analysis.

On the other hand, next-generation sequencing (NGS) uses short-read technologies, producing significantly smaller fragments lengths, in a way greatly enhanced the coverage at the expense of intensive computational power [111]. The fully automated NGS is conceptually similar to shotgun sequencing. However, the approach used differs from one method to another, resulting to multitudes of analytical performances as illustrated in Table 5.

**Table 5.** Sequencing platforms of various generations and the throughputs and turnaround times [112,113].

| Method | Gen. | Read Length (bp) | No. of Reads per Run | Error Rate per Run (%) | Read Time |
|---|---|---|---|---|---|
| Sanger 3730×1 | 1st | 600–1000 | 96 | 0.001 | 0.3–3 h |
| Ion Torrent semiconductor | 2nd | 200 | $8.2 \times 10^7$ | 1 | 2–4 h |
| Roche 454 Pyrosequencing | 2nd | 700 | $1 \times 10^6$ | 1 | 1 d |
| Illumina HiSeq 3000/4000 (High throughput) | 2nd | $2 \times 150$ | $8 \times 10^9$ (paired) | 0.1 | 1–4 d |
| SOLiD 5500×1 | 2nd | $2 \times 60$ | $8 \times 10^8$ | 5 | 6 d |
| PacBio RSII SMRT | 3rd | ~$1.0–1.5 \times 10^4$ | $3.5–7.5 \times 10^4$ | 13 | 0.5–4 h |
| Oxford Nanopore MinION | 3rd | $2–5 \times 10^3$ | $1.1–4.7 \times 10^4$ | 38 | 2 d |

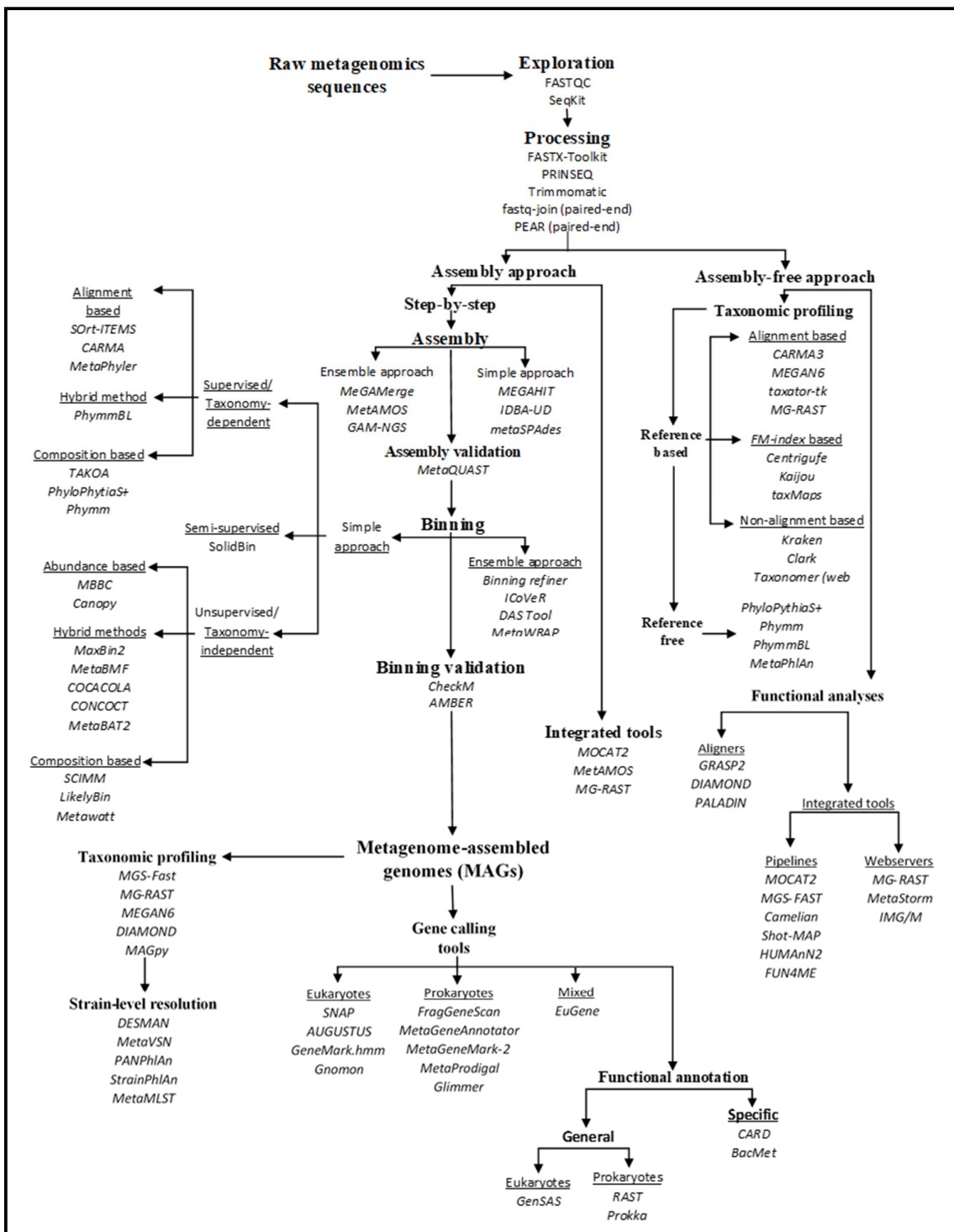

**Figure 8.** Schematic flow showing necessary steps for the analysis of metagenomics raw data. Bioinformatic programs relevant to each stage are listed in italics [110].

Third generation NGS completely bypassed the clonal amplifications step, giving the benefit of time and cost, while simultaneously allowing direct sequencing of gDNA [114]. Initially, such an approach is limited to only those projects working on microbial genomes. However as more computational power is acquired, such as from the development of MinHash Alignment Process (MHAP) and BLASR for noise reduction in long reads, significantly larger genomes such as human and invertebrate genomes can also be studied [115].

## 4. Conclusions

This review intends to fill the gaps between different disciplines involved in the efforts to limit the impacts of hydrocarbon pollutants in the Antarctic continent and the extended marine territories. The scientometric analyses can help polar scientists in grasping the past and current progresses made, relevant to diesel problems in Antarctica. Based on the analyses, the relevance of authors' research interests (as indicated by keywords selection) has been proven to be in-sync with the current trend in Antarctic studies. This analysis also highlighted several emerging fields of studies, shedding light on future possibilities in the bioremediation of oil pollutants (Table 6).

**Table 6.** Field clusters and some of their prospects based on research keywords in relevance to hydrocarbons pollution.

| Cluster | Prospects |
| --- | --- |
| Remediation strategies | Discovery of autochthonous hydrocarbonoclastic microorganisms; development of immediate response frameworks towards hydrocarbon pollutions; optimization of existing clean-up methods currently being employed in Antarctica. |
| Sustainability | Repurposing of general wastes in Antarctic stations and proper managements; partial incorporation of more renewable power sources for daily routines; safeguarding of both scientific and public interests (biota and wilderness areas) in Antarctica. |
| Anthropogenic impacts and management | Revision of transportation routes throughout Antarctic regions to minimize foreign contaminations by vessels; revamping or bolstering of existing laws in the Annex of Environmental Protocol to the Antarctic Treaty; establishment of proactive approaches that limit the capacity of tourists per annum as well as the type of tourisms. |

However, some limitations encountered in this study are worth noting such that the source(s) of data may be biased and limited to certain indexed publications. Secondly, the selection of a query string will affect the whole outcomes of the analysis in which too many inclusions or omittance of keywords can mislead the authors into making erroneous analysis.

**Author Contributions:** Conceptualization, A.Z., S.A.A. and A.F.A.R.; Writing—original draft preparation, A.F.A.R.; Software, A.F.A.R.; Writing—review and editing, A.Z., S.A.A., C.G.-F., N.A.S., and K.A.K.; Supervision, A.Z., S.A.A., C.G.-F., N.A.S., and K.A.K. All authors have read and agree to the published version of the manuscript.

**Funding:** This project was financially supported by Shibaura Institute of Technology. The authors also thank the Public Service Department of Malaysia (JPA) for granting a personal scholarship (PhD programme) to Ahmad Fareez Ahmad Roslee.

**Institutional Review Board Statement:** Not applicable.

**Informed Consent Statement:** Not applicable.

**Data Availability Statement:** No new data were created or analyzed in this study.

**Acknowledgments:** The authors would like to thank Shibaura Institute of Technology, Universiti Putra Malaysia (GPM-9678900), Centro de Investigacion y Monitoreo Ambiental Antàrctico (CIMAA) and Sultan Mizan Antarctic Research Foundation (YPASM-6300247).

**Conflicts of Interest:** The authors declare no conflict of interest.

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
