# Peer review of "Scientometric Analysis of Diesel Pollutions in Antarctic Territories: A Review of Causes and Potential Bioremediation Approaches"

_sustainability, doi:10.3390/su13137064_

Round 1
Reviewer 1 Report
In conclusion topic:"This analysis also highlighted several emerging field of studies, shedding light to future possibilities in the bioremediation of oil pollutants" - I suggest a pragraph or a table with the main emerging field of studies and their respective possibilities, to make it them clear, as the text is extense. Maybe authors can create this table immediatily before the conclusion topic.
Author Response
Answer:
The emerging fields of studies can be referred through Figure 6 and Figure 7, by which we have identified according to the research clusters and keywords occurrence in recent years.
As for the future possibilities, some of them were mentioned throughout the texts. Anyhow, a brief summary (Table 6) was added in the conclusion section. Page 22-23.
Reviewer 2 Report
Dear Authors,
you have performed good research on the topic and analyzed obtained literature giving interesting and valuable review.
Good job
Author Response
Thank you for your supportive comments
Reviewer 3 Report
Greetings for the authors,
The paper under review is titled "Scientometric analysis of diesel pollutions in Antarctic territories: A review of causes and potential bioremediation approaches".
The article is well organized and theme is relevant for the society.
I just have few things to suggest.
There is something wrong with the page numbering. Please check it.
Verify the citations of the Figures and Tables inside the text. They are not following the journal rules.
Replace "petrogenic hydrocarbons" with "thermogenic hydrocarbons". Biogenic hydrocarbons can also be generated inside the deep petroleum reservoirs (rocks).
PAH means "polyciclic aromatic hydrocarbons", and MAH means "monocyclic aromatic hydrocarbons". Please, fix it.
The numering (2.6) of the subchapter "Genes Families Conferring to Hydrocarbons..." is clearly wrong. Please, fix it.
In the "Introduction", the part between "In order to generate meaningful visual representation..." and "...distributions by countries and subjects." should be in methodology chapter. The introduction needs more state of art and geographical context.
I hope it helps.
Author Response
Comment 1: There is something wrong with the page numbering. Please check it. Answer: Corrected, there might be technical error during the editing process.
Comment 2: Verify the citations of the Figures and Tables inside the text. They are not following the journal rules. Answer: We went through the template provided online, and it seems that the way we cited the table/figure complied to the rules set by the journal. Added ( ) for sentence that ends with Figure/Table.
Comment 3: Replace "petrogenic hydrocarbons" with "thermogenic hydrocarbons". Biogenic hydrocarbons can also be generated inside the deep petroleum reservoirs (rocks). Answer: Replaced the term petrogenic with thermogenic where applicable in the manuscript. Page 4.
Comment 4: PAH means "polyciclic aromatic hydrocarbons", and MAH means "monocyclic aromatic hydrocarbons". Please, fix it. Answer: Corrected as per comment. Thank you. Page 10.
Comment 5: The numering (2.6) of the subchapter "Genes Families Conferring to Hydrocarbons..." is clearly wrong. Please, fix it. Answer: Corrected numbering to subchapter 3.5. Page 14.
Comment 6: In the "Introduction", the part between "In order to generate meaningful visual representation..." and "...distributions by countries and subjects." should be in methodology chapter. The introduction needs more state of art and geographical context. Answer: Subchapter 3.3 explains on how the post-processing (co-occurrence and manual clustering) of collected data were done using built-in function using VosViewer software. Chapter 3 was written in a way that it consists of the detailed methods used during the scientometric analysis followed by the interpretation of the results. In contrast, Chapter 2 (methodology) largely explains on how the data were gathered prior to this study. Due to the nature of this review, details regarding state of art and geographical context are discussed in subchapter 3.4.